# Feeding Behavior Responses of the Small Copepod, *Paracalanus parvus*, to Toxic Algae at Different Concentrations

**DOI:** 10.3390/ani13193116

**Published:** 2023-10-06

**Authors:** Zixuan Ding, Xiaohong Sun, Yiming Qiao, Ying Liu, Jihua Liu

**Affiliations:** 1Marine College, Shandong University, Weihai 264209, China; sisyphus990403@163.com (Z.D.); 15610373627@163.com (Y.Q.); 2Institute of Marine Science and Technology, Shandong University, Qingdao 266237, China; liujihua1982@foxmail.com; 3Weihai Marine and Fishery Monitoring and Hazard Migration Centre, Weihai 264209, China; whmemc@126.com

**Keywords:** toxic dinoflagellates, high-speed camera, copepods, feeding behavior

## Abstract

**Simple Summary:**

The feeding relationship between copepods and phytoplankton has immense ecological significance. The feeding behavior of copepods may be affected by the ingestion of toxic algae. This study investigated the feeding behavior of copepods via study of the feeding selectivity of *Paracalanus parvus*, a key small copepod species, using a high-speed camera. The study demonstrates that toxic dinoflagellates alter the feeding behavior of copepods and describes the variations in their feeding response to different algal species and concentrations. The findings provide crucial insights for further studies on the feeding relationship between copepods and phytoplankton and the functional assessment of plankton ecosystems.

**Abstract:**

The feeding relationship between copepods and phytoplankton has immense ecological significance. This study investigated the feeding behavior of copepods by studying the feeding selectivity of *Paracalanus parvus*, a key small copepod species, using a high-speed camera. The feeding behavior of *P. parvus* separately fed on three algae, *Prorocentrum minimum, Alexandrium minutum*, and *Thalassiosira weissflogii*, was studied at five different concentrations. The factors characterizing feeding behavior, including the beating frequency (BF), beating time (BT), and rejection behavior, were analyzed. The average BT and BF of *P. parvus* fed on toxic algae were significantly lower than those of copepods fed on nontoxic algae, indicating that the toxic algae negatively affected their feeding behavior. There were no significant differences in feed rejection among the three algae during the short period of experimentation, indicating that the rejection behavior was insignificant in the early period (within 20 min) of feeding on toxic algae. The feeding behavior was inhibited when the concentration reached 250 cells/mL. The BT was initially affected at increasing concentrations followed by the BF, and *P. minimum* and *A*. *minutum* reduced the BF at concentrations of 250 and 1000 cells/mL, respectively. Analysis of the average BFs revealed that *P. parvus* was more significantly affected by *P. minimum* containing diarrheal shellfish poison than by *A*. *minutum* containing paralytic shellfish poison. The BF of copepods fed on *P. minimum* was significantly lower than that of copepods fed on *A*. *minutum* at 250–500 cells/mL but was not significantly different from that at 1000 cells/mL. This indicated that the inhibitory effect of *P. minimum* on the feeding behavior was more significant at concentrations observed at the onset of red tide blooms (0.25–0.5 × 10^2^ cells/mL), but insignificant at concentrations reaching those in advanced red tides (>10^3^ cells/mL). This study demonstrates that toxic dinoflagellates alter the feeding behavior of copepods and describes the variations in their feeding response to different algal species and concentrations. The findings provide crucial insights for further studies on the feeding relationship between copepods and phytoplankton and on functional assessment of plankton ecosystems.

## 1. Introduction

Copepods are one of the most abundant groups of marine zooplankton [1] and act as important links in marine ecosystems. They can regulate phytoplankton populations and communities via top-down control as well as via bottom-up control mechanisms by serving as direct or indirect nutrient sources for larvae and juvenile fishes. Previous studies have demonstrated that the feeding habits of copepods play an important role in their population dynamics and have a highly significant role in mediating the transfer of matter and energy in marine ecosystems [2]. Their feeding habits also play an important role in regulating the mechanism of red tide outbreaks via top-down control approaches. Phytoplankton constitute the main food of copepods. The″ black box″ method, which indirectly calculates the difference in algal concentration before and after feeding, has been widely used to study the feeding rates of copepods and their preference for phytoplankton. However, it has been reported that the black box method produces a bias for copepods that feed on small algae [3,4]. The effects of toxic algae on copepods have been studied using this method [5], and harmful algae have various negative effects on copepods [6], including the malformation of eggs and larvae, reduction of fecundity in adults, and retardation of development [7,8,9]. However, the findings only describe group behaviors and do not provide insights into specific behavioral changes in individuals [5,10]. Additionally, the mechanism underlying the feeding behavior response of copepods to different concentrations of toxic algae remains to be elucidated.

Therefore, the feeding preference and responses of copepods to toxic algae need to be determined using direct methods. Studies on the microscopic behaviors of copepods have developed rapidly in recent years owing to the use of high-speed cameras in zooplankton research.

Kiørboe et al. classified the feeding behaviors of copepods into three categories, namely, ambush, cruising, and feeding-current types, and provided a theoretical framework for further elucidation of their feeding mechanisms [10,11]. The method used by Kiørboe et al. has been applied for studying the escape velocity [12] and ingestion of microplastics by copepods [13,14]. Xu et al. investigated the effects of toxic algae on copepods using a high-speed camera and directly observed the feeding behaviors of copepods. The results indicated that the beating activity of the feeding appendages (including BT and BF) reduced by ca. 80% during the initial 60 min of exposure, which was accompanied by specialized behaviors such as rejection and regurgitation [5,10]. It is therefore necessary to investigate whether different species of toxic algae, such as those containing diarrheal shellfish poison (DSP) or paralytic shellfish poison (PSP), have varying effects on the feeding behaviors of copepods. It is also necessary to determine the concentration at which toxic algae affect the feeding behavior of copepods in toxic red tide blooms.

*Paracalanus* spp. belong to the order Calanoida, and Paracalanidae are the typical dominant copepods in coastal waters [15,16]. The feeding behavior of *Paracalanus spp.* from different domains has been studied using traditional feeding methods; the results have demonstrated that the copepods in this genus are herbivorous and have a relatively high feeding rate on heterotrophic dinoflagellates [17]. Of these, *P. parvus* is a common dominant species in the coastal waters of China [18,19]. As a small copepod (< 1 mm), *P. parvus* plays an important role in linking the microbial food loop with classical food chains [20]. Studies on the feeding habits of *P. parvus* in the Bohai Sea revealed that their food primarily consists of diatoms and dinoflagellates, which comprise 99.6% and 0.4%, respectively, of their food, and *Coscinodiscus* spp. are the dominant diatom species in the region [21]. However, the results were obtained by studying the anatomy of the digestive tract, which may have introduced some bias, as dinoflagellates are more easily digested than diatoms, which have thick silicon shells [17]. It has been reported that *Paracalanus* spp. have a relatively high feeding rate on heterotrophic dinoflagellates; they also feed on dinoflagellate species in red tides, including *Alexandrium excavatum* and *Prorocentrum micans*, and have relatively high filtration rates, but the rate of filtration decreases at increasing algal concentrations [22,23].

Toxic dinoflagellate blooms have become a serious ecological disaster worldwide, including in the coastal waters of China [24,25,26], and several studies have reported the presence of toxic dinoflagellates, including *Prorocentrum minimum* and *Alexandrium minutum*, in algal blooms [24,27]. It is therefore necessary to elucidate the feeding behavior responses of copepods, especially the dominant small copepod, *P. parvus*, to toxic dinoflagellate blooms.

*P. parvus* is a feeding-current type copepod, and the BF and BT of the feeding appendages are important parameters for characterizing the active feeding behavior of copepods. Under normal conditions, the feeding appendages beat nearly constantly to produce a feeding current; however, the beating activity of the feeding appendages reduces when there is a shortage of food supply. It remains to be determined whether the feeding behavior of *P. parvus* alters in the presence of varying algal concentrations. The present study aimed to investigate the feeding behavior of *P. parvus* using a high-speed camera by studying its feeding ability, especially on the dinoflagellates in red tides at different concentrations. We hypothesized that *P. parvus* can filter different species of phytoplankton at different concentrations by altering its feeding behavior.

In this study, the feeding behavior responses of the important dominant small copepod, *P. parvus*, to toxic and harmful phytoplankton were studied using a high-speed camera. The study predicted that *P. parvus* would change its feeding behavior after feeding on toxic algae, indicated by a reduction in the BF and BT of the feeding appendages and the appearance of rejection behavior. We also predicted that the reduction in the BT and BF of the feeding appendages and the rejection rate would be more pronounced at increasing algal concentrations. The findings provide direct insights into the effects of feeding on toxic algal species in small copepods and the mechanism underlying their response to toxic algae in red tides.

## 2. Materials and Methods

### 2.1. Experimental Organisms

Three algal species were used for the experiments in this study, including two toxic dinoflagellates, namely, *P. minimum* and *A*. *minutum*, containing diarrheal shellfish poison (DSP) and paralytic shellfish poison (PSP), respectively, and one non-toxic diatom, *Thalassiosira weissflogii*. The three algae were provided by Shanghai Guangyu Biological Technology Co., Ltd. and cultured at 20 °C, under a light intensity of 2500 lux and a 12 h/12 h light/dark cycle. Before experimentation, the algae were counted and diluted with filtered seawater to desired concentrations of ~50, ~100, ~250, ~500, and ~1000 cells/mL.

The feeding behavior of *P. parvus*, a dominant species of zooplankton in the Yellow Sea, was investigated in this study. Samples were collected on a daily basis during sunset from 22 July to 1 August 2022. The sampling site was located in the Shuangdao Bay of Weihai, at the northern part of the Yellow Sea (121 58′ 45.135″ E, 37 28′ 30.017″ N), and the samples were collected at a depth of approximately 20 m. Multiple zooplankton samples were collected from the bottom and brought to the surface using a shallow-water Type II plankton net (mesh: 160 μm, diameter: 37 cm). The captured zooplankton was quickly placed in a 37.5 L Coleman thermos cabinet and 15 L seawater was added. After sampling, the samples of zooplankton were transported to the laboratory for processing within two hours. The samples of healthy *P. parvus* were sorted and placed in a beaker with filtered seawater. The sorted copepods were domesticated for 2 hours in a thermo-constant room at 20 °C.

### 2.2. Behavioral Observations

The feeding behavior of *P. parvus* was recorded during the grazing period using a high-speed camera. The three algal species were separately provided to *P. parvus* at five different concentrations of ~50, ~100, ~250, ~500, and ~1000 cells/mL. Each treatment setup consisted of six individual *P. parvus* added to a cuvette (1 cm × 1 cm × 5 cm) with the corresponding concentrations of algal seawater.

The feeding behavior of *P. parvus* was recorded using a FASTCAM NOVA S16 high-speed camera with infrared illumination incident through the cuvette and toward the camera, which was equipped with NAVIT AR lenses, and each device was mounted on an optical porous plate for stability. The high-speed camera was adjusted for recording the feeding behavior of *P. parvus* (frame rate: 1000 Hz; resolution: 768 × 512 pixels). Several 2.0–3.0 s sequences of recording were obtained for each of the treatment groups for observing the capture, ingestion, and rejection events during feeding and for quantifying the factors characterizing feeding behavior. The feeding behavior of the copepods in each of the treatment groups was recorded within 20 min of providing the algal feed. The changes in three feeding factors, namely, the BF, beating time (BT), and feed rejection, were recorded and analyzed as described hereafter. The BF was defined as the frequency of beating of the feeding appendages of the copepods within 1 ms during feeding, and the BT was defined as the duration of beating of the feeding appendages of the copepods in one feeding event. Feed rejection was defined as the rejection of feed during the beating of the feeding appendages when feeding on algal prey.

### 2.3. Statistical Analyses

Videos of the feeding behavior of *P. parvus* were analyzed and processed using the PFV4 software. The differences in the feeding factors among different prey species and concentrations were determined by one-way analysis of variance (ANOVA). The assumption of homogeneity of variances was assessed by Levene′s test. The homogeneity of variance was determined by the square root transformation method when this assumption was violated. Tukey′s test was used to compare the means for multiple comparisons. The results are presented as the mean ± standard deviation, and statistical significance was considered at *p* < 0.05.

## 3. Results

### 3.1. Feeding Behavior of P. parvus on Different Concentrations of P. minimum

The changes in the BF of *P. parvus* in the presence of different concentrations of *P. minimum* feed are depicted in Figure 1. The highest BF of 121.94 ± 1.82 Hz was observed when the concentration of *P. minimum* was 100 cells/mL, and the BF did not differ significantly from that at a concentration of 50 cells/mL (*p* = 0.20), but was significantly higher than those at concentrations of 250–1000 cells/mL (*p* < 0.01). These findings indicated that the BF decreased significantly when the concentration of *P. minimum* increased beyond 250 cells/mL. The BT ranged from 0.28 ± 0.03 s–0.38 ± 0.06 s across the different concentrations of *P. minimum* tested herein, and there were no significant differences in the BT when the concentration of *P. minimum* was altered (*p* > 0.05 at all concentrations). This finding indicates that the BT of *P. parvus* was independent of the concentration of *P. minimum* and was determined to be 0.34 ± 0.02 s on average.

These results indicate that the BF of *P. parvus* was relatively high (108.65 ± 1.88–121.94 ± 1.82 Hz) when the concentrations of *P. minimum* were 50 and 100 cells/mL, but it was affected when the concentration of *P. minimum* was higher than 250 cells/mL.

### 3.2. Feeding Behavior of P. parvus Fed on Different Concentrations of A. minutum

In Figure 2, the highest BF of 124.06 ± 1.23 Hz was observed when the concentration of *A*. *minutum* was 100 cells/mL and was higher than those at concentrations of 50 and 250 cells/mL (*p* < 0.01 and *p* < 0.05, respectively), but did not differ significantly from the BFs at concentrations of 250 and 500 cells/mL. The lowest BF of 106.96 ± 2.39 Hz was observed when the algal concentration was 1000 cells/mL, and differed significantly from the BFs at concentrations of 100–500 cells/mL (*p* < 0.01). These results demonstrate that the BF of *P. parvus* increased when the concentration of *A*. *minutum* increased from 50 to 100 cells/mL. However, the BF did not differ significantly when the algal concentration was further increased from 100 to 500 cells/mL, but decreased significantly when the concentration of *A*. *minutum* was increased to 1000 cells/mL. The highest BT of 0.43 ± 0.028 s was observed when the concentration of *A*. *minutum* was 100 cells/mL, and was significantly higher than the lowest value of 0.21 ± 0.025 s observed at a concentration of 50 cells/mL (*p* < 0.05). Compared to that at 100 cells/mL, the BT decreased significantly when the concentration of *A*. *minutum* increased to 250 cells/mL (*p* < 0.01), but increased when the concentration was further increased to 500–1000 cells/mL (0.28 ± 0.02 s).

These results demonstrate that both the BF and BT of *P. parvus* increased when the concentration of *A*. *minutum* increased from 50 to 100 cells/mL, indicating that the feeding rate of *P. parvus* increased at lower concentrations of *A*. *minutum* (50–100 cells/mL). However, the feeding rate of *P. parvus* was negatively affected by higher concentrations of *A*. *minutum*, which was indicated by the reduction in BT at a concentration of 250 cells/mL, and the apparent reduction in BF when the concentration of *A*. *minutum* reached those observed in red tide blooms (1000 cells/mL).

### 3.3. Feeding Behavior of P. parvus Fed on Different Concentrations of T. weissflogii

In Figure 3, the highest BF of 124.93 ± 1.06 Hz was observed when the concentration of *T. weissflogii* was 500 cells/mL, which did not differ significantly from the BF at a concentration of 1000 cells/mL (*p* = 0.07), but was higher than those observed at concentrations of 100 and 250 cells/mL (*p* < 0.01). The BF did not differ significantly at algal concentrations ranging between 50 and 100 cells/mL or between 100 and 250 cells/mL (*p* > 0.05 at all concentrations); however, the BF at a concentration of 50 cells/ml differed significantly from that at a concentration of 250 cells/mL (*p* = 0.04). These results demonstrated that the BF of *P. parvus* remained relatively low until the concentration of *T. weissflogii* increased to 500 cells/mL, and the BF remained high even at a concentration of 1000 cells/mL. The BT of the feeding appendages of *P. parvus* ranged from 0.34 ± 0.20–0.46 ± 0.30s across all concentrations of *T. weissflogii* tested herein, and there were no significant differences in the BT across the different concentration (*p* > 0.05). These findings indicate that the BT of *P. parvus* remained stable when fed on *T. weissflogii* and did not alter significantly when the concentration of the prey was increased. 

These results indicate that the BT and BF of *P. parvus* were not affected when fed on a high concentration of *T. weissflogii*, and decreased under higher algal concentrations. This indicates that *P. parvus* exhibited a more active feeding behavior when fed on the algae *T. weissflogii*, even when its concentration reached those observed in red tides (500–1000 cells/mL).

### 3.4. Comparison of the Feeding Behavior of P. parvus Fed on Different Concentrations of Three Algae

On average, the BF and BT of *P. parvus* fed on the nontoxic diatom were significantly higher than those in the presence of the two toxic dinoflagellates in Figure 4. Comparison of the BFs of *P. parvus* fed on the two toxic dinoflagellates revealed that the BF in the presence of *P. minimum* was significantly lower than that in the presence of *A*. *minutum* feed. However, there was no statistical difference in the BTs when fed on the two toxic dinoflagellates.

The feeding behavior of *P. parvus* in the presence of the two toxic algae was first compared with that in the presence of the same concentration of the nontoxic *T. weissflogii*. Comparison of the BF and BT of *P. parvus* fed on low concentrations (50–100 cells/mL) of *P. minimum* and *T. weissflogii* revealed no significant differences in the BF and BT (*p* > 0.05). However, the BT of *P. parvus* fed on *A*. *minutum* was significantly higher than in the presence of *T. weissflogii* fed at a concentration of 100 cells/mL. These results indicate that the feeding behavior of *P. parvus* was not negatively affected by the toxic dinoflagellates at low concentrations (<100 cells/mL).

The BT of *P. parvus* fed on the toxic algae was significantly lower than that in the presence of *T. weissflogii* at the same concentration of 250 cells/ml (*p* < 0.01). The BF of *P. parvus* fed on the toxic algae was significantly lower than that in the presence of *T. weissflogii* at concentrations of 500–1000 cells/ml. Altogether, these findings indicate that the feeding behavior of *P. parvus* was not significantly affected by low concentrations (50–100 cells/mL) of the two toxic dinoflagellates. However, the toxic algae affected the feeding behavior of *P. parvus* at high concentrations (≥ 250 cells/mL); the effects of higher algal concentrations were mainly reflected in the BT of *P. parvus*, and a further increase in the algal concentration primarily affected the BF. The BT of *P. parvus* in the presence of high concentrations (500–1000 cells/mL) of the toxic dinoflagellates did not differ significantly from that in the presence of the same concentration of the nontoxic diatom, which led us to speculate that *P. parvus* had to maintain the normal duration of filter-feeding owing to the demand for food following the obvious reduction in BF.

However, the average BF of *P. parvus* fed on *P. minimum* was significantly lower than that of copepods fed on *A*. *minutum* at both concentrations (*p* < 0.01), and there was no significant difference between the two toxic algae in terms of the BF at a concentration of 1000 cells/mL. Comparison of the BT of *P. parvus* fed separately on *P. minimum* and *A*. *minutum* revealed that there were no significant differences in the BT at higher algal concentrations; however, the BT of copepods fed on *A*. *minutum* was lower than that of *P. parvus* fed on *P. minimum* at the lowest concentration of 50 cells/mL tested herein. These results indicate that there were no significant differences in the feeding behavior of *P. parvus* when fed on low concentrations of *P. minimum* or *A*. *minutum*. However, the effects of *P. minimum* were more pronounced than those of *A*. *minutum* at higher concentrations of 250–500 cells/mL. The changes were mostly evident in the BF of the appendages, and both the toxic algae markedly altered the BF of *P. parvus* when their concentration increased to 1000 cells/mL. 

The significance of the differences in all data is shown in Table 1

### 3.5. Comparison of Rejection Behavior of P. parvus Fed on Different Concentrations of the Three Algae

The feed rejection percentages of *P. parvus* separately fed on the three algal species are provided in Table 2. The average rejection percentage of *P. minimum* feed was 7.95%, and the highest rejection percentage of 13.51% was observed at a concentration of 1000 cells/mL. The average rejection percentage of *A*. *minutum* was 8.15%, and the highest rejection percentage of 17.86% was observed at a concentration of 500 cells/mL. The rejection percentage of *T. weissflogii* was lowest, being 4.83% on average, and the highest feed rejection percentage of 13.64% was observed at a concentration of 500 cells/mL. The results of statistical analyses revealed that prey rejection was independent of the concentration of the prey for all the three algal species tested. The findings further revealed that there were no significant differences among the three algae in terms of the rejection percentage (*p* > 0.05), indicating that rejection was not an important factor for determining the effects of toxic algae on the feeding behavior of *P. parvus*. 

Figure 5 shows the complete process of rejection of algal particles by *P. parvus* under high-speed camera.

## 4. Discussion

### 4.1. Effects of Toxic Algae on the Feeding Behavior of Copepods

The two toxic algae selected herein, namely, *P. minimum* and *A*. *minutum*, have a wide global distribution [28,29], and their blooms cause massive economic losses and serious health hazards for humans. The most significant human mortality to date was caused by a *P. minimum* bloom containing DSP in Lake Hamana, Japan in March 1942, causing fatality in 114 of the 324 affected individuals following the consumption of oysters (*Venerupis semidecussata*) and short-necked clams (*Tapes semidecussata*) [30]. Another outbreak of *P. minimum* and subsequent events of poisoning have been frequently reported in several individuals in Massachusetts, Alaska, and Norway [31,32]. The presence of PSP in marine red tides with *A*. *minutum* as a typical species is responsible for the frequent poisoning events and has a serious effect on human health [33,34]. Apart from the effects on human health, PSP can directly damage the gills and mantle tissues of shellfish, as these tissues respond first to toxic dinoflagellates [35]. DSP and PSP not only have toxic effects on humans, shellfish, and other species, but also affect copepods, which are the primary predators of toxic phytoplankton.

Using a single feeding experiment, a previous study demonstrated that copepods are less likely to ingest toxic algae than nontoxic algae of similar size [36]. The feeding behavior responses of copepods to *Alexandrium*. spp. have been previously investigated and the findings revealed that their feeding behavior alters in the presence of toxic algae, indicated by the reduction in feeding and feeding rate [7]. The results of the present study demonstrated that the BF and BT of *P. parvus* fed on the nontoxic *T. weissflogii* were significantly higher than those of *P. parvus* fed on the two toxic algae (Figure 4, average data), indicating that the two toxic algae negatively affected the feeding behavior of *P. parvus*. The results of the present study on the feeding behavior of the small copepod, *P. parvus*, are consistent with the findings of previous studies and indicate that toxic algae can suppress the feeding behavior of copepods.

The effects of feeding on the two toxic algal species were further compared, and the findings revealed that the BF of *P. parvus* fed on *P. minimum* was significantly lower than that of the copepods fed on *A*. *minutum* (Figure 4, average data); however, there were no significant differences in the BT, which indicated that the inhibitory effect of *P. minimum* on the feeding behavior of *P. parvus* was more pronounced than that of *A*. *minutum*. A previous study demonstrated that the feeding rate of the copepod *Temora longicornis* fed on *A*. *minutum* is higher than that of copepods fed on *P*. *lima*, which also contains DSP similar to *P. minimum* [37], thus confirming that the inhibitory effect of *Prorocentrum.* spp. on the feeding behavior of copepods could be more pronounced than that of *A*. *minutum*. *P. minimum* is a dominant dinoflagellate species in China and is responsible for causing several red tide events in the North Yellow Sea and East China Sea [38,39,40]; its cysts have been found on the seafloor in China [41,42]. DSP is one of the most common algal toxins in the offshore regions of China, and is known to contaminate shellfish, having caused several poisoning events [24,27]. Therefore, further studies are necessary for determining the effects of *P. minimum* on the feeding behavior of the more dominant copepods, their ecological effects, and the differences between the effects of DSP and PSP.

### 4.2. Effects of High Concentrations of Toxic Algae on the Feeding Behavior of Copepods

Apart from the algal toxins, the cellular concentration of toxic algae is also a key factor in determining the feeding behavior of copepods. Turrif et al. performed copepod feeding experiments using *Alexandrium tamarind* and *T. weissflogii*, and the results demonstrated that the toxic *A*. *tamarind* algae negatively affected the feeding behavior of copepods at increasing concentrations, which was not observed in the presence of *T. weissflogii* [43]. The present study similarly demonstrated that the feeding behavior of *P. parvus* was not negatively affected by *T. weissflogii*; however, the toxic dinoflagellates had obvious negative effects on the feeding behavior at increasing concentrations.

The present study initially demonstrated that the BF and BT were not negatively affected by *T. weissflogii* at increasing concentrations (Figure 3). The findings also revealed that, starting at 250 cells/mL, both the toxic algae, *P. minimum* and *A. minutum*, negatively affected the feeding behavior of *P. parvus* as the concentrations increased. Moreover, the results demonstrated that the BT first decreased significantly when the concentration of *A. minutum* reached 250 cells/mL, and the obvious negative effects on the BF of the appendages were observed at 1000 cells/mL, when the concentration of *A. minutum* reached those observed in red tides.

Comparison of the effects of the two toxic algae on the feeding behavior of *P. parvus* with that of the same concentration of *T. weissflogii* revealed that the toxic algae negatively affected the feeding behavior of *P. parvus* (Figure 4, Table 2). The BT decreased markedly in the presence of *P. minimum* and *A*. *minutum*, when the concentrations increased to 250 cells/mL. These results, together with the finding that the toxic algae reduced the BF at concentrations of 500–1000 cells/mL, confirmed that the toxic dinoflagellates did not have any obvious negative effects on the feeding behavior of copepods at low concentrations. The toxic effects were first evident on the BT at a concentration of 250 cells/mL, and then on the BF at concentrations ≥ 500 cells/mL. There were no significant differences among the three algae in terms of the BT of the appendages at concentrations higher than 500 cells/mL. This was possibly attributed to the fact that *P. parvus* reduced the BF of the appendages and maintained a relatively normal BT to conserve energy and obtain the most basic nutrients required for sustaining life under high concentrations of toxic algae.

Altogether, these results demonstrate that both the toxic algae negatively affected the feeding behavior of the copepod *P. parvus* at increasing concentrations, and revealed that the nontoxic diatom, *T. weissflogii*, did not have any negative effects on the feeding behavior of *P. parvus*. Contrary to the results obtained in this study, the findings of the study by Xu et al. [10] revealed that the variations in the BF of the feeding appendages were independent of the prey concentration (40–200 cells/mL) and time of feeding (within 4 hours). This discrepancy could be attributed to the relatively low concentrations of algae used in the study by Xu et al. In the present study, the algal feed was provided at concentrations of 50–1000 cells/mL, ranging from normal concentrations in seawater to that in red tide blooms. The results also confirmed that the toxic dinoflagellates negatively affected the feeding behaviors of *P. parvus* at concentrations higher than 250 cells/mL.

As discussed in Section 4.1, the inhibitory effect of *P. minimum* on the feeding behavior of *P. parvus* was more pronounced than *A*. *minutum*, and we further compared their negative effects at increasing concentrations. The BFs of *P. parvus* fed on *P. minimum* and *A*. *minutum* were similar at low concentrations (50–100 cells/mL, *p* (50) = 0.067, *p* (100) = 0.408). The BT of *P. parvus* fed on *P. minimum* was significantly higher than that of the copepods fed on *A*. *minutum* at the same concentration of 50 cells/mL (Figure 4, Table 2, *p* < 0.01), indicating that there were no obvious differences between the negative effects of the toxic algae at low concentrations. However, the toxic algae had varying effects on the feeding behavior at higher concentrations of 250–500 cells/mL, in that the BF of *P. parvus* fed on *P. minimum* was significantly lower than that of the copepods fed on *A*. *minutum*. Additionally, there were no significant differences in the BT of *P. parvus* at both concentrations, indicating that the negative effect of *P. minimum* on the feeding behavior of *P. parvus* was more pronounced at high concentrations (250–500 cells/mL). However, there were no significant differences between the toxic algae at a concentration of 1000 cells/mL in terms of the BF of the appendages, indicating that both algae could affect the feeding behavior of copepods at high concentrations. These results suggest that studies aimed at investigating the negative effects of toxic algae on the feeding behavior of copepods need to use algal feed at concentrations higher than 250 cells/mL.

### 4.3. Mechanisms Underlying the Alterations in Feeding Activity

Based on the findings of our study on the effects of toxic algae on the feeding behavior of *P. parvus*, we further investigated the mechanism underlying the effects of toxic algae on the feeding behavior of copepods. Some studies suggest that copepods use their sensing and detecting ability to actively resist toxic algae [43,44,45,46]. Analysis of the high-speed videos acquired in this study revealed that all the active rejection events were mediated by the feeding appendages when they touched the toxic algal cells. Statistical analyses revealed that the average rejection rates of *P. parvus* were 4.83%–8.15% for the three algal species, and there were no significant differences among the three algae in terms of the rejection rate (Table 2). These results indicate that copepods have the ability to actively avoid toxic algae, but the alterations in the feeding behavior of *P. parvus* to toxic algae could be attributed to another potential selective mechanism.

Xu et al. investigated the feeding behavior of the filter-feeding copepod, *Te*. *longicornis*, and the results demonstrated the copepod exhibited four distinct feeding behavioral responses to different algal prey, namely, (i) normal feeding behavior; (ii) significant reduction in appendage BT in the first hour of introducing the algal prey—the BT remained low thereafter and most of the captured cells were ingested, although at a low rate; (iii) regurgitation—appendage beating remained high and the captured algal cells were ingested at a high rate, followed by subsequent regurgitation of a large fraction of the ingested material; and (iv) feeding activity and rate of prey capture remained high, but increasing fractions of captured cells were rejected during the first hour, while the rejection rate remained high during the remainder of the observation period [10]. The first two modes of feeding were also observed in the present study. In this study, the″ normal″ feeding behavior was characterized by the nearly constant beating of the feeding appendages, which produced a feeding current, and most (90%) of the captured algae were ingested, confirming that the rejection percentage was less than 10%. The second mode of feeding was also observed in this study, which was characterized by the reduction in the BT and beating activity of the feeding appendages. The findings reveal that both the BF and BT of *P. parvus* fed on the two toxic algae were significantly lower than those of copepods fed on the nontoxic diatom (Table 2, average data). The third mode of feeding (regurgitation) was not observed in *P. parvus* and the results could not be compared with the findings on *Te*. *longicornis*. The study by Xu et al. reported that, in the fourth feeding behavior, the increase in the fraction of *A*. *pseudogonyaulax* rejected by *Te*. *longicornis* was independent of the concentration of the algal prey (40–200 cells/mL), and the majority of the captured cells were rejected after 60 min. The findings of the present study also confirmed that the rejection rates of *P. parvus* were independent of the prey concentration for all the three algal species tested herein; however, as the videos on copepod feeding were acquired over a period within 20 min of introducing the feed, the rejection rate after 60 min of feeding could not be analyzed.

The findings of other studies have suggested that the changes in the feeding behavior of copepods in response to toxic algae could be attributed to the ingested algal poison, which results in slower movement and reduces the food intake [43,47,48,49,50]. Xu et al. suggested that this could be possibly mediated by substances released during food processing in the gut, as the reduction in feeding activity is only observed after the ingestion of a few algal cells. The videos captured within 10 min of introducing the feed using a high-speed camera revealed that the BF and BT of *P. parvus* fed on the two toxic algae were significantly lower than those of copepods fed on the nontoxic diatom (Table 2, mean value at all concentrations). These findings indicate that the feeding behavior of filter-feeding copepods decreases in response to toxic algae within a short time of feeding.

## 5. Conclusions

In this study, the feeding behavior of *P. parvus* was observed using a high-speed camera for investigating whether toxic algae affect the feeding behavior of copepods, and whether this effect varies according to the concentration of different toxins. The results demonstrated that the feeding behavior of the copepod *P. parvus* was inhibited by the toxic algae, *P. minimum* and *A*. *minutum*, compared to that of copepods fed on the nontoxic *T. weissflogii*, and the inhibitory effect became more pronounced at increasing concentrations. It was further observed that *P. minimum*, which contains DSP, had a more pronounced effect on the feeding behavior of *P. parvus* than *A*. *minutum*, which contains PSP. The findings revealed that microalgae containing DSP had more significant negative effects on the feeding behavior of *P. parvus* than microalgae containing PSP. Comparison of the rejection behavior of *P. parvus* revealed that there were no significant differences among the three algae in terms of the rejection rate, indicating that active rejection is not the primary mechanism underlying the identification of toxic algae by *P. parvus*. The findings of the present study further revealed that the BF and BT of the feeding appendages are reduced in response to the ingestion of algal toxins.

This study aimed to identify the negative effects of toxic algae on the feeding behavior of *P. parvus* from the perspective of individual behavior by analyzing videos acquired using a high-speed camera. The findings provide a theoretical basis and methodological guidance for subsequent studies on copepod feeding behavior and management of red tides in the North Yellow Sea.

## Figures and Tables

**Figure 1 animals-13-03116-f001:**
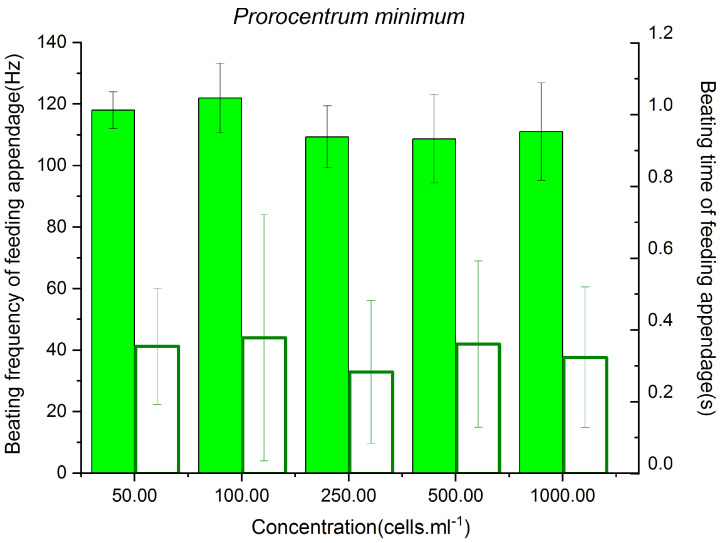
BF and BT of the feeding appendages of *P. parvus* fed on different concentrations of *A*. *minutum*. The BF is indicated by solid green bars (left y-axis; unit: Hertz), and the BT is depicted by white bars (right y-axis; unit: second).

**Figure 2 animals-13-03116-f002:**
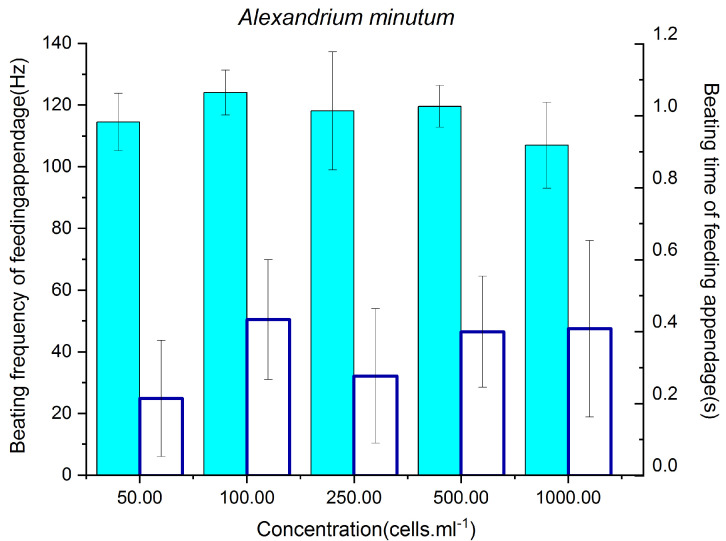
BF and BT of the feeding appendages of *P. parvus* fed on different concentrations of *A*. *minutum*. The BF of the feeding appendages are depicted by light blue bars (left y-axis; unit: Hertz), whereas the BT is depicted by white bars (right y-axis; unit: second).

**Figure 3 animals-13-03116-f003:**
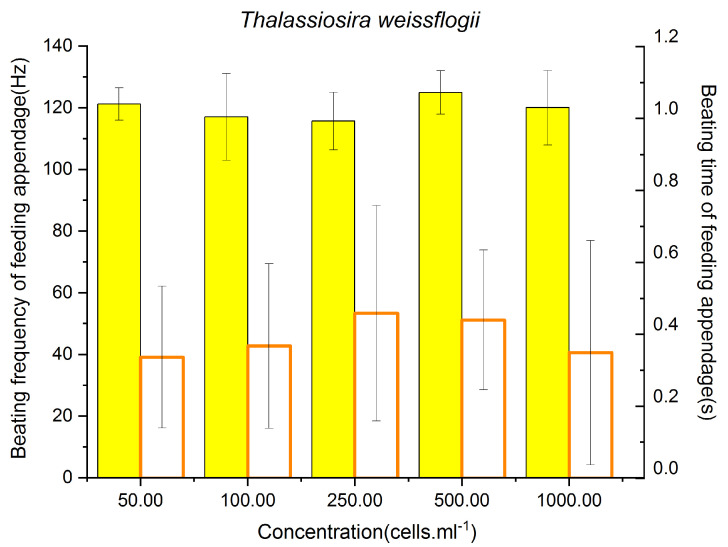
BF and BT of the feeding appendages of *P. parvus* fed on different concentrations of *T. weissflogii*. The BF is indicated by the yellow bars (left y-axis; unit: Hertz), and the BT is represented by the white bars (right y-axis; unit: second).

**Figure 4 animals-13-03116-f004:**
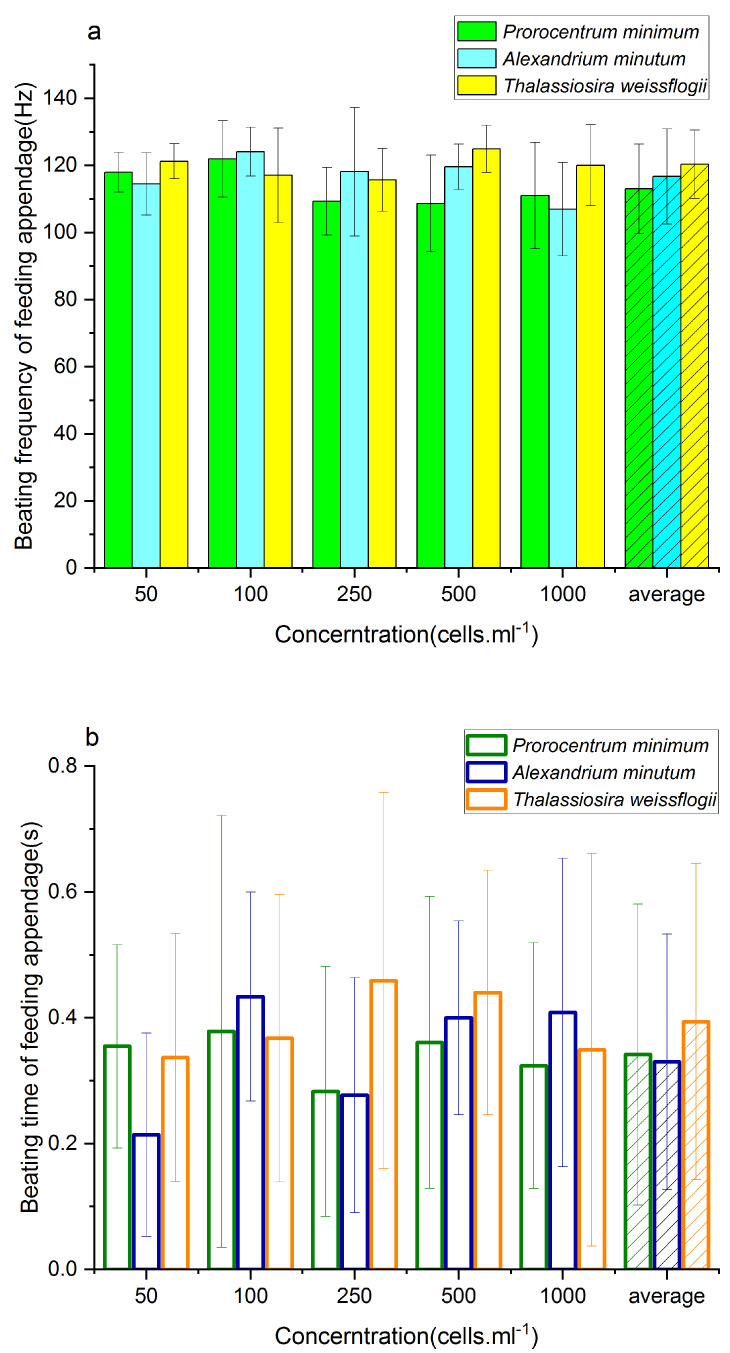
Feeding behavior of *P. parvus* fed on different concentrations of the three algal species. (**a**) BF and (**b**) BT of the feeding appendages.

**Figure 5 animals-13-03116-f005:**
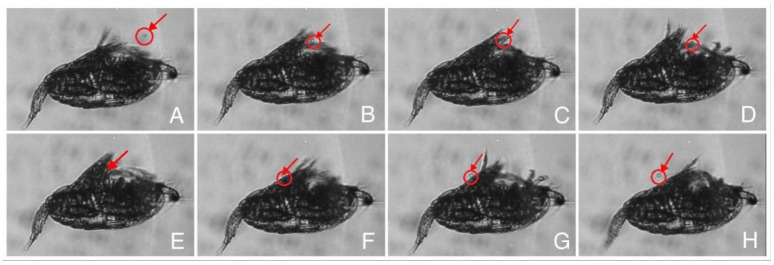
Process of prey rejection by *P. parvus*.

**Table 1 animals-13-03116-t001:** Results of statistical analyses of the data obtained for each algal species at different concentrations.

Species			*p* (Concentration in Cells/mL)					
			*p* (50)	*p* (100)	*p* (250)	*p* (500)	*p* (1000)	*p* (Average)
BF of feeding appendage	*P. minimum*	*T. weissflogii*	0.10	0.08	0.08	<0.01	<0.01	<0.01
	*A*. *minutum*	*T. weissflogii*	<0.01	<0.01	0.47	<0.05	<0.01	<0.01
	*P. minimum*	*A*. *minutum*	0.07	0.41	<0.01	<0.01	0.23	<0.01
BT of feeding appendage	*P. minimum*	*T. weissflogii*	0.70	0.87	<0.01	0.06	0.67	<0.05
	*A*. *minutum*	*T. weissflogii*	<0.01	0.34	<0.01	0.43	0.34	<0.01
	*P. minimum*	*A*. *minutum*	<0.01	0.37	0.90	0.41	0.17	0.64
Rejection percentage	*P. minimum*	*T. weissflogii*						0.35
	*A*. *minutum*	*T. weissflogii*						0.32
	*P. minimum*	*A*. *minutum*						0.95

**Table 2 animals-13-03116-t002:** Rejection percentage as determined from the videos of *P. parvus* fed on different concentrations of the three algae.

Concentration	Percentage of Feed Rejection					
	*P. minimum*		*A*. *minutum*		*T. weissflogii*	
50 cells/mL	10.71%	n = 26	2.38%	n = 42	3.33%	n = 30
100 cells/mL	6.82%	n = 39	8.57%	n = 35	3.85%	n = 26
250 cells/mL	5.26%	n = 38	6.67%	n = 60	3.33%	n = 20
500 cells/mL	3.45%	n = 58	17.86%	n = 27	13.60%	n = 44
1000 cells/mL	13.51%	n = 37	5.26%	n = 34	0.00%	n = 35
Average	7.95%		8.15%		4.83%	
Std. dev.	4.10%		5.88%		5.14%	

n represents the total number of videos analyzed for different concentrations of each algal species.

## Data Availability

Data available on request due to privacy issues or ethical restrictions.

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
