# Peer review of "Feeding Behavior Responses of the Small Copepod, Paracalanus parvus, to Toxic Algae at Different Concentrations"

_animals, 2023, doi:10.3390/ani13193116_

Round 1

Reviewer 1 Report

I think it is an interesting study, but the paper can be improved by clearly specifying in the introduction the relationship between BF and BT and feeding behavior. You also need to clearly state what is expected for normal feeding behavior what what is expected for feeding behavior that is "negatively" influenced. Additionally, there are mistakes e.g. the figure 4 is not a comparison of all algae tested, but a duplication of the non-toxic one. I believe that you can reduce the discussion by summarizing your results in comparison with other studies rather than going through each "test" one-by-one. There are also mistakes and I believe unnecessary references in the reference list. Please see comments, changes, etc. in the attached .pdf file.

Your English needs some attention. I found it strange that the English in most of the paper needs attention, except for the first paragraph of the Introduction which is almost faultless. 

Author Response

Dear reviewer,I am honored to receive your professional advice and recommendations,I read your advice carefully,first,I revised the introduction to explain the behavior of copepods during normal feeding and possible changes in feeding behavior during abnormal feeding, emphasizing the role and importance of parameters in the article,Secondly, I have tried to revise parts of the article to reduce the lengthy fluff, modified the images and tables, and cut out unnecessary references and standardized the format of the references. As for changes and clarifications for specific paragraphs, I have addressed and labeled them accordingly in the article, so thank you again for your help in our work!

Reviewer 2 Report

The manuscript presents interesting results of the feeding selectivity of Paracalanus parvus using a high-speed camera. This species frequently predominates among copepods in coastal waters. The research showed a small impact of different algae species and different densities on the feeding behavior of P. parvus. Nevertheless, the negative effects of toxic algae on the feeding behavior of P. parvus were observed. These findings provide valuable insights into copepod feeding behavior and possible management of red tides in the North Yellow Sea (and others).

A few comments (more in the ms):

The English language should be improved before publication.

There are some repetitions in the Introduction and Discussion. Some part of the discussion is a description of the results.

Please correct and adjust the references according to the journal's standard (e.g., many are written in capital letters).

Please edit the Figures – there is a lack of space between the words (x and y axis). Please consider showing significant differences on the graph (different letters or *). Box plots rather than columns are much better for presenting descriptive statistics (Figures 1-4). Figure 4 is the same as Figure 3.

Discussion in lines 407-437 (and 447-464) is a description of results without reference to the literature.

More comments and corrections are in the manuscript file.

The manuscript is generally well written, however the English language should be improved before publication. Please correct and adjust the references according to the journal's standard.

Author Response

Dear reviewer,I am honored to receive your professional advice and recommendations,I read your advice carefully,first,I have deleted the repetition of lengthy fragments in the article, but in the last paragraph of the Discussion 4.3 section we have chosen to keep it as we believe that here is an explanation and elaboration of the later text, which is so closely related to it that it may be detrimental to the flow of the article if it is not briefly clarified.Secondly, I have modified the images and tables, and cut out unnecessary references and standardized the format of the references. In addition, we appreciate your suggestion to change the format of the picture, but we think that Figure 4 has more data, and if we change it to a box plot and then mark it with significance, it will look cluttered, and the first three figures show the same parameters as in Figure 4, and we think that we should standardize the format, so we prefer to keep the original format, and the results of the significance analysis are listed in the table. I wonder if you agree with us? As for the English expression part of the article, I try to make changes to the expression of the article, and if it is still substandard, we will continue to make changes.As for changes and clarifications for specific paragraphs, I have addressed and labeled them accordingly in the article, so thank you again for your help in our work!

Reviewer 3 Report

Review for the paper "Feeding behavior responses of the small copepod, Paracalanus parvus, to toxic algae at different concentrations" by Zixuan Ding, Xiaohong Sun, Yiming Qiao, Jihua Liu,Ying Liu submitted to "Animals".

General comment.

Most predictions have documented that harmful algae blooms would induce strong changes in coastal marine ecosystems. Copepods, being the most numerous zooplankton groups worldwide, represent good organisms to study possible impact from toxic algae on the health of the inshore pelagic communities. It is expected that toxicants released with microalgae will have a strong influence on the copepod metabolism, reproduction, and behavior. The objective of the present study was to analyze behavioral responses of a common copepod Paracalanus parvus on toxic microalgae. The paper is well structured and has a good visualization of the main results. Discussion is comprehensive. However, there are many linguistic errors and uncertainties in the ms that should be corrected in the revised version.

Specific remarks.

L12. Paracalanus parvus must be in italics.

L12. capitalize “we”.

L12. Consider replacing "demonstrates" with "demonstrate".

L13. Consider replacing "describes" with "describe".

L36-40. It is suggested to omit these sentences from the Abstract.

L49-51, 54-56. Provide relevant references.

L70. Delete "of assessment".

L89, 100, 104 onward. Consider replacing " Pa. " with "Paracalanus".

L93. Consider replacing " micro " with "microbial".

L91-97 and L108-112 are the same text. Delete the repetition.

L115 onward. Consider replacing " A. sp " with " Alexandrium sp.".

L124-142. The text contains some repetitions and should be corrected.

Fig. 4 is the same as Fig. 3. Please, provide relevant Fig. 4 to show your results.

L333. Consider replacing " in Table 1. " with " in Table 2. ".

Section 5. There is no reference to Fig. 5. Please, correct.

Reference list. The authors must carefully check the references. In most cases, the Latin names of species and genera are not italicized. Please, correct. Also, the bibliographic citations in the “References” section were a sloppy mess. In some cases journal titles were abbreviated (properly) but in other cases titles were completely spelled out. In some cases only initial words and proper names in titles were capitalized, whereas in other reference titles, all nouns were capitalized.

1. The main research questions are:

How did the toxical microalgal concentrations affect feeding behavior of a common copepod?

Were there significant changes in behavioral responses on different food concentrations?

2. The topic is original only from the point of view of the research area (sampling region for Paracalanus). Other findings are mainly repetitive. There has been a bulk of similar studies since 2000-s. The authors’ results in general contribute little to the current knowledge.

3. The study provides new data regarding rejection rate of microalgae. The authors provided high-quality photos to illustrate the processes. However, considering no effect of microalgal concentrations, this result has a little significance in the field.

4. The authors sampled the experimental species in one seasons (summer) and location. Copepods are known to have different physiological conditions in relation to season and sampling site. One possible suggestion for authors to compare Paracalanus individuals caught from different locations and seasons.

5. But can be improved by comparing with other copepod taxa.

6. May be significantly improved by adding recent papers.

English revisions are required.

Author Response

Dear reviewer,I am honored to receive your professional advice and recommendations,I would also like to thank you for recognizing and supporting our work, which encourages us.I read your advice carefully,In response to the comments and questions you gave, I offer the following answers:

  1. The main research questions are:

How did the toxical microalgal concentrations affect feeding behavior of a common copepod?

Were there significant changes in behavioral responses on different food concentrations?

Answer: Filter-feeding copepods will reduce feeding on toxic algae through detection and recognition, and produce changes in feeding behavior such as a decrease in the frequency of appendage beating and a decrease in the duration of feeding after feeding, and this effect will be enhanced with increasing algal concentrations, but the effect varies depending on the species of copepods and the size of the individuals.

  1. The topic is original only from the point of view of the research area (sampling region for Paracalanus). Other findings are mainly repetitive. There has been a bulk of similar studies since 2000-s. The authors’ results in general contribute little to the current knowledge.

Answer: We attempted to utilize coexisting copepods and algae in the sea area for our study, tending to target specific ecological phenomena. Admittedly, many researchers have contributed to the study of copepod feeding behavior, but this experiment used high-speed camera technology to visualize copepod feeding behavior and reflect the effect of algae on copepods from an individual perspective. More importantly, we allowed the copepods to swim naturally in the water column and captured their feeding on camera instead of utilizing the tethering method as other researchers have done, which makes our data closer to the natural situation.

  1. The study provides new data regarding rejection rate of microalgae. The authors provided high-quality photos to illustrate the processes. However, considering no effect of microalgal concentrations, this result has a little significance in the field.

Answer: Experiments on rejection rates were conducted in parallel with experiments on the observation of feeding behavior, and the data included the rejection rates of copepods when feeding on different concentrations of the three algae. We took into account the concentration effect factor and presented data on rejection rates at different concentrations. Specific data are presented in Table 2.

  1. The authors sampled the experimental species in one seasons (summer) and location. Copepods are known to have different physiological conditions in relation to season and sampling site. One possible suggestion for authors to compare Paracalanus individuals caught from different locations and seasons.

Answer: Thank you for your suggestion, we will consider it for future studies.

  1. But can be improved by comparing with other copepod taxa.

Answer: Thanks to your suggestion, we have followed up with a comparison of the feeding behavior of large copepods with that of small copepods, and the article is in progress.

  1. May be significantly improved by adding recent papers.

Answer: Thanks to your suggestion, we have revised the references to include several articles after 2010.

Round 2

Reviewer 2 Report

The manuscript seems to be revised and looks better. However, the response to the Reviewers is not clear (which reviewer? Replies without previous comments; I do not find responses to many of my comments, etc.)

There are still many small errors (e.g., lines: 20, 97, 313, 350, 387, etc.) Please check carefully whole text.

Please use the same number of decimal places. Sometimes 4 decimal places it’s too much, etc.

The references were not corrected and adjusted to the journal's format.

The language requires correction. There are still many small errors.

Author Response

Dear Reviewer, It is great to receive your feedback again. I apologize for increasing your workload because I didn't clearly identify my response to your suggestion the first time, this time, I have revised the decimal places as well as the formatting of the references, and I have explained and revised your previous suggestion in the article, but I am sorry that I didn't clearly see the error in line 20, could you please point it out for me? Secondly, this article has been touched up by a professional, and the touch-up record is attached below for you, if you still feel that it is not up to standard, we will make further changes. Thank you again for your guidance.
